# m6A-Dependent RNA Dynamics in T Cell Differentiation

**DOI:** 10.3390/genes10010028

**Published:** 2019-01-08

**Authors:** Mattia Furlan, Eugenia Galeota, Stefano de Pretis, Michele Caselle, Mattia Pelizzola

**Affiliations:** 1Center for Genomic Science, Fondazione Istituto Italiano di Tecnologia, 20139 Milan, Italy; mattia.furlan@iit.it (M.F.); eugenia.galeota@iit.it (E.G.); stefano.depretis@iit.it (S.d.P.); 2Physics Department and INFN, University of Turin, 10125 Turin, Italy; caselle@to.infn.it

**Keywords:** RNA-seq, RNA dynamics, m6A, RNA modifications, RNA metabolic labeling, mathematical modeling

## Abstract

N6-methyladenosine (m6A) is the most abundant RNA modification. It has been involved in the regulation of RNA metabolism, including degradation and translation, in both physiological and disease conditions. A recent study showed that m6A-mediated degradation of key transcripts also plays a role in the control of T cells homeostasis and IL-7 induced differentiation. We re-analyzed the omics data from that study and, through the integrative analysis of total and nascent RNA-seq data, we were able to comprehensively quantify T cells RNA dynamics and how these are affected by m6A depletion. In addition to the expected impact on RNA degradation, we revealed a broader effect of m6A on RNA dynamics, which included the alteration of RNA synthesis and processing. Altogether, the combined action of m6A on all major steps of the RNA life-cycle closely re-capitulated the observed changes in the abundance of premature and mature RNA species. Ultimately, our re-analysis extended the findings of the initial study, focused on RNA stability, and proposed a yet unappreciated role for m6A in RNA synthesis and processing dynamics.

## 1. Introduction

Dozens of RNA modifications decorating coding and non-coding RNA species have been shown to be important determinants in the fate of marked RNA transcripts [1]. m6A is the most abundant RNA modification in various species, including human and mouse. Its pattern is determined by the action of specific writers (including METTL3) and erasers, and impacts on most stages of the RNA life-cycle, including processing, stability, transport, and translation [2]. Regulation by m6A is involved in various important biological processes such as cellular differentiation, pluripotency, stress response, and gametogenesis [3,4,5,6]. Moreover, alterations in m6A patterns are associated with human diseases like cancer initiation and progression [7,8,9]. While the impact of m6A on specific stages of the RNA life-cycle, such as RNA translation and degradation, is supported by ample evidence and widely recognized. Its involvement in other steps, namely RNA synthesis and processing, is far from being fully established. However, both a number of components of the m6A writer complex and the erasers are enriched within nuclear speckles, suggesting their interplay with the machineries responsible for RNA transcription and processing. Indeed, m6A installment was shown to occur prevalently at the level of premature RNA [10] and to be influenced by RNA Polymerase II elongation [11]. In addition, few studies have indicated a role for m6A in the regulation of alternative splicing [12,13,14,15,16,17], but opposite evidence has also been reported [10,18].

Recently, an RNA metabolic labeling method has been developed that detects nascent RNA molecules based on the incorporation of modified 4sU nucleotides. By allowing the quantification of the kinetic rates governing the various stages of the RNA life-cycle, this method permits to characterize the RNA dynamics and therefore, in our case, to investigate how the m6A epitranscriptome impacts on the dynamics of premature and mature RNA species.

A recent study combining RNA metabolic labeling with m6A profiling demonstrated the relevance of m6A in the control of T cell homeostasis and IL-7 induced differentiation [19]. Its authors analyzed the impact of m6A on RNA degradation both in wild type (WT) T cells and in T cells in which m6A writer *Mettl3* had been knocked-out (KO) thus leading to a marked reduction in m6A bulk levels (28% of WT levels). When key transcripts involved in T cell homeostasis and differentiation were considered, m6A was found to heavily influence their stability while having a negligible role in their processing and translation

We re-analyzed the RNA metabolic labeling data from the same study. To this end, we quantified the kinetic rates of RNA synthesis, processing and degradation using INSPEcT (version 1.8.0), a tool that we had previously developed for the integrative analysis of nascent and total RNA-seq data [20]. Our analyses suggest that in T cells, the m6A epitranscriptome is important, not only for the control of RNA degradation, but also in RNA synthesis and processing.

## 2. Materials and Methods

### 2.1. Dataset Description

Raw RNA-seq data on total and nascent RNA and raw data for m6A profiling in untreated WT cells were derived from the GEO series GSE100278 [19], where WT and *Mettl3*-KO mouse naive T cells were compared before and after IL-7 stimulation. Nascent RNA was quantified by RNA metabolic labeling with a 15 min pulse of 4sU nucleotides. In WT cells, total and nascent RNA were quantified before IL-7 induction, and at 15, 30, 45, 60, and 75 min following IL-7 exposure. In *Mettl3*-KO cells, total and nascent RNA were quantified in untreated cells, and after 15, 30, and 60 min of exposure to IL-7.

We focused our analysis on the samples obtained from untreated cells and from cells exposed for 15 and 60 min to IL-7 treatment, as these were available for both WT and KO cells. We ignored the samples from 30 min exposure since, in this case, the WT samples gave low numbers of aligned reads (0.9 and 2 million reads respectively, compared to an average of 1.7 and 5.6 million aligned reads in other samples) and a low correlation with other samples (Appendix A).

### 2.2. Analysis of High-Throughput Sequencing Data

Raw data were processed using HTS-flow, a workflow management system for high-throughput sequencing data that we had previously developed [21]. Briefly: (i) poor quality bases were masked, (ii) reads were aligned with the mouse genome (mm9 assembly) using TopHat with default settings [22], (iii) aligned reads were filtered for duplicates using the samtools rmdup routine [23].

m6A peaks were obtained using the MACS peak caller (version 2.0) with a *p*-value cutoff of 1 × 10^−5^. m6A+ transcripts were identified by requiring at least one MACS peak within each of the two available replicates. m6A− transcripts were identified as those never associated to MACS peaks. Only genes with adequate intronic and exonic expression were retained (see Section 2.3).

### 2.3. Expression Quantification and Filtering

BAM alignment files were analyzed using the R/Bioconductor package INSPEcT [20]. The abundance of premature RNA species was determined based on the intronic signal from total RNA-seq experiments. That of mature RNA species was determined by subtracting the intronic signal from the exonic signal. Reads densities within these genomic regions were quantified using a function in INSPEcT called makeRPKMs (based on transcripts annotation from the TxDb.Mmusculus.UCSC.mm9.knownGene Bioconductor package (version 3.2.2). Intron-less genes and genes with either exonic expression lower than 1 RPKM (Reads Per Kilobase of transcript, per Million mapped reads) or intronic expression lower than 0.1 RPKM in at least one sample, were excluded. The remaining 4527 genes underwent further analysis.

### 2.4. Mathematical Modeling of Synthesis, Processing, and Degradation Rates

The time-courses of total and nascent RNA-seq data in WT and KO conditions were analyzed with INSPEcT. Due to lack of biological replicates, it was not possible to estimate mean values and variances for the expression levels of the genes. This issue forced us to limit our analysis to first guess rates estimates, which were not subjected to any modeling step. However, first guess and modeled rates are typically well correlated. For example, in Reference [20] the median Pearson’s correlations between first guess and modeled rates were 0.89, 0.80, and 0.72 for synthesis, processing, and degradation, respectively.

A fraction of the processing and degradation computed rates were not finite (processing WT: 0.9%, processing KO: 0.4%, degradation WT: 20.8%, degradation KO: 11.9%). In these cases, we evaluated the missing values by using a linear model to interpolate the rates computed immediately before and after the not finite datum. When this situation occurred at the boundary of the time-course, we returned the closest finite value.

### 2.5. Analysis of Data Distributions

The distributions of the changes in RNA kinetic rates in Figure 1B were tested for a shift from 0 using the non-parametric Wilcoxon test. In particular, due to the large difference in the number of data points within each distribution, the *p*-values were determined by sampling the all and m6A− distributions based on the size of the m6A+ population. The median *p*-value over 1000 repetitions was reported in the figure.

The distributions of premature and mature RNA abundances and of the RNA kinetic rates in Figure 2 were compared between WT and KO time-courses. To avoid making assumptions on the normality of the distributions, the Kolmogorov–Smirnov (KS) and the Mann–Whitney (MW) tests were used to assess how significant differences were.

### 2.6. Functional Enrichment Analysis

Functional enrichments of regulated transcripts were performed using the R/Bioconductor package rGREAT version 1.11.1 [24], based on the genes corresponding to the 500 transcripts with the highest log2 fold change in KO compared to WT, in any of the kinetic rates. The following ontologies and categories were considered: GeneOntology (GO) Molecular Function, GO Biological Process, GO Cellular Component, Mouse Phenotype, Disease Ontology, PANTHER Pathway, BioCyc Pathway, and MSigDB Pathway. Individual terms were considered significant when both *p*-values of the hypergeometric and binomial tests were below the threshold of 1 × 10^−3^. Only *p*-values of the hypergeometric test were finally reported. The same approach was used to analyze the functional enrichment of gene clusters.

### 2.7. Clustering 

Cluster membership was determined based on the abundance of premature and mature RNA species and on the RNA kinetic rates. Specifically, we considered the absolute values in the untreated condition and the changes that followed IL-7 treatment in WT and *Mettl3*-KO cells.

The absolute values of mature RNA species were quantified as Z-scores from the joint distribution of untreated WT and KO data by subtracting the median and normalizing over the interquartile value. The same procedure was applied to premature RNA abundances, synthesis rate, processing rate, and degradation rate data.

The changes that followed IL-7 treatment were quantified by computing the log2 ratios at 15 and 60 min compared to the untreated condition for premature and mature RNA species and RNA kinetic rates, in both WT and KO cells.

We verified that Z-scores and fold changes had comparable medians and interquartile ranges (Appendix A). We then clustered the matrix composed of Z-scores and fold changes in 11 groups of genes using the k-means algorithm (R package stats version 3.4.1) based on a standard sum of squares metric.

Various number of groups were tested in the range between 2 and 15, and 11 groups were chosen as a trade-off between: (i) minimizing the percentage of clusters with less than 25 genes; (ii) maximizing uniqueness and statistical significance of the Biological Processes related to each cluster. Regarding the second task, we performed a functional enrichment analysis based on the GO Biological Process ontology for each clustering configuration. For all groups of clusters, we selected the statistically significant terms (hypergeometric *p*-value < 1 × 10^−5^) and estimated their mean *p*-value and the median percentage of terms that appeared only in one cluster (Appendix A).

A simplified version of the resulting heatmap is discussed in Results—Clusters analysis. Specifically, it does not include clusters 9–11, because represented by less than 25 genes, it does not contain the 15 min log2 fold changes and for each couple of Z-scores (WT and KO), it reports only a per gene mean value. The complete heatmap is displayed in Appendix A.

### 2.8. Source Code

The R source code for reproducing all the analyses is available in Appendix A.

## 3. Results

### 3.1. Global Consequences of m6A Depletion on T Cells RNA Dynamics

In order to characterize the impact of m6A depletion on RNA dynamics, we used INSPEcT to quantify the rates of RNA synthesis, processing, and degradation in both *Mettl3*-KO and WT T cells (Figure 1A). We then compared their distributions for 4527 transcripts expressed with adequate levels in both their mature and premature forms. In agreement with the original study [19], the rates of degradation were significantly reduced following m6A decrease (Figure 1B). Indeed, in KO naive T cells the transcripts showed a global increase in stability. Interestingly, synthesis and processing rates were also reduced in KO cells (Figure 1B). Taking advantage of the m6A profiling within untreated WT cells, we confirmed these trends on the subset of m6A+ transcripts. Noteworthy, the same trend could be observed also for m6A− transcripts, while with a lower significance (Figure 1B). 

We used functional enrichment analysis to characterize the transcripts with the largest changes in RNA kinetic rates (Figure 1C). Transcripts with lower degradation rates in KO cells were not enriched for specific terms, suggesting that the overall reduction in degradation was a broad consequence of m6A depletion and did not preferentially affect any specific class of transcripts. On the contrary, transcripts with the opposite pattern, i.e., higher degradation rates in KO cells were involved in protein and nucleic acid metabolism (metabolic process, *p* = 7 × 10^−9^, and nitrogen compound metabolic process, *p* = 1.0 × 10^−4^). Transcripts with higher processing rates in KO cells were enriched in terms related to energy production (mitochondrion, *p* = 2 × 10^−9^, and Genes involved in Respiratory electron transportation, *p* = 4 × 10^−5^), while those with lower processing rates were related to RNA translation, processing, and metabolism (translation, *p* = 5 × 10^−8^ “nucleic acid metabolic process, *p* = 3 × 10^−11^ and spliceosomal complex, *p* = 1 × 10^−6^). Noteworthy, although m6A depletion did not globally affect the rates of synthesis, transcripts with increased synthesis rates in KO cells were enriched in immune system and cytokine response (response to cytokine stimulus, *p* = 2 × 10^−5^, and Genes involved in Immune System, *p* = 2 × 10^−5^).

Altogether these analyses indicate that, in unstimulated T cells, m6A exerts a global control on the stability of transcripts. They also suggest that m6A plays a role in determining synthesis and processing dynamics, specifically for classes of transcripts involving co- and post-transcriptional regulation. It remains to be confirmed whether these effects are direct or indirect consequences of m6A.

### 3.2. Impact of m6A Depletion on T Cells Treated with IL-7

We characterized the effect of m6A depletion on T cells by comparing RNA kinetic rates in WT and *Mettl3*-KO T cells, at 15 and 60 min after induction with IL-7 (further details on why these specific time-points were selected can be found in Materials and Methods—Dataset description). In WT cells, all kinetic rates became affected after 15 min of IL-7 treatment (Figure 2A).

After 60 min, these effects stayed unchanged in the case of synthesis and processing rates, while increased with regard to degradation rates. Only synthesis and degradation rates were globally affected in KO cells, while processing rates were mostly unaffected (Figure 2B). Notably, while synthesis rates increased in both WT and KO cells, their induction was delayed in KO cells. Additionally, while degradation rates increased in both WT and KO cells, this effect was partially reversed following 60 min of IL-7 induction in KO cells. These observations suggest an m6A− dependent response to IL-7 induction in terms of: (i) timing in the induction of synthesis rates, (ii) modulation in the processing rates, and (iii) sustained up-regulation of degradation rates.

### 3.3. Clusters Analysis

Cluster analysis was used to identify groups of genes with coordinated response to IL-7 in terms of RNA species abundance and RNA kinetic rates (Figure 3).

Clusters 1 and 2 were the largest clusters (composed of 1496 and 1853 genes, respectively), grouping genes spanning across various functional categories, including: RNA metabolic process, protein metabolic process, chromatin modification, catalytic activity and binding (*p* < 1 × 10^−10^). These clusters showed similar inductions in terms of synthesis and premature RNA, but different behaviors in terms of total RNA. In fact, following IL-7 treatment in both WT and KO cells, the total RNA abundance decreased for cluster 1 genes while increased for cluster 2 genes. Moreover, these two clusters showed very different patterns in terms of degradation rates. Cluster 1 was characterized by stable transcripts of which half-lives were markedly reduced after IL-7 induction. On the contrary, cluster 2 was characterized by less stable transcripts whose degradation rates partially increased. Noteworthy, the increase in the degradation rates of cluster 2 transcripts was dampened in KO cells, suggesting that m6A depletion partially impaired post-transcriptional regulation.

Cluster 3 (composed of 93 genes) included highly expressed genes characterized by fast kinetics both in terms of synthesis and degradation rates. Following IL-7 induction, despite a reduction in their rate of synthesis, the total RNA abundance of these transcripts increased due to an increase in RNA stability, which seemed to be more homogeneous in m6A depleted cells. These transcripts were maintained at high level of expression despite high degradation rates, denoting the investment of a significant amount of energy. Due to their low stability, these transcripts were fast responders in case of perturbations and could be expected to encode critical functions in T cells. Indeed, these genes were involved in biological processes related to abnormal T cell differentiation (*p* = 5 × 10^−13^), and to RNA processing (RNA splicing, *p* = 4 × 10^−9^, and mRNA processing, *p* = 2 × 10^−9^).

Similar to cluster 3, transcripts in cluster 5 (composed of 410 genes) were highly expressed and synthesized. The corresponding genes were involved in hematologic diseases (hematologic cancer, *p* = 4 × 10^−12^, and hematopoietic system disease, *p* = 2 × 10^−11^). Differently from cluster 3, cluster 5 transcripts were stable. IL-7 induction resulted in an increase in their degradation, leading to their down-regulation. Cluster 7 (composed of 53 genes) included highly expressed genes that were quickly synthesized and slowly degraded, and which differed from cluster 5 transcripts for their extremely high processing rates. In WT cells, IL-7 induction caused a remarkable decrease in the processing rates of the genes in cluster 7, together with a strong decrease in their stability. *Mettl*-3 KO cells showed a similar trend but with responses of noticeably smaller magnitudes, highlighting then how the regulation of the processing and degradation kinetics depends on m6A. The genes forming cluster 7 were mainly involved in translation (*p* = 5 × 10^−35^).

Finally, cluster 8 (composed of 251 genes) was characterized by the highest processing rates, which showed a decrease in response to IL-7 dampened by m6A depletion. As expected, the modulation of processing did not affect total RNA levels, while it originated the observed increase in premature RNA abundance. Interestingly, genes were enriched for terms involving RNA processing (RNA processing, *p* = 2 × 10^−9^, mRNA processing, *p* = 5 × 10^−8^, spliceosomal complex, *p* = 2 × 10^−8^, Genes involved in Processing of Capped Intron-Containing Pre-mRNA, *p* = 2 × 10^−9^). This suggests a feedback regulatory strategy, where transcripts involved in RNA processing modulate their own processing dynamics.

Altogether, the characterization of the effects of m6A depletion on the IL-7 response reinforces the patterns described for the untreated condition (Figure 1), suggesting a role for m6A not only in the regulation of degradation dynamics but also in the control of synthesis and processing dynamics. Cluster analyses illustrate how the combined modulation of various RNA kinetic rates closely explain and re-capitulate the observed transcriptional modulation for both premature and mature RNA species.

## 4. Discussion

Similar to other RNA modifications, dynamic m6A marks have the potential to impact various stages of an RNA life-cycle. Nonetheless, the involvement of m6A has been mostly elucidated in relation to the degradation and translation of transcripts. Indeed, the impact of m6A on transcriptional and co-transcriptional regulation, including RNA synthesis and processing, is far from being established. In particular, the impact of m6A on the dynamics of these fundamental cellular processes is mostly unknown.

A study was recently published that investigated the impact of m6A depletion on the dynamics of premature and mature RNA species in T cells [19]. In that study, the authors focused on m6A-dependent RNA degradation and showed its importance in the case of a few genes that are crucial to T cells differentiation. Our genome-wide analysis re-capitulate and broaden the impact of m6A on RNA degradation in T cells. In addition, we suggest that m6A marks affect other stages of the RNA life-cycle. In the context of T cells homeostasis, m6A depletion causes a global slowdown for all the RNA kinetic rates. Moreover, it impacts on all RNA kinetic rates during T cells differentiation, by: (i) delaying the induction of synthesis rates, and impairing both (ii) the modulation of processing rates, and (iii) the sustained up-regulation of degradation rates. Noteworthily, these effects might be direct or indirect consequences of m6A and its deregulation. Further experiments are necessary to investigate the prevalence of these two alternative scenarios.

Overall, our analysis suggests a broader impact of m6A on the dynamics of the RNA life-cycle and highlights the importance of studying the role of m6A marks in combination with experiments able to profile the full set of RNA kinetic rates. We anticipate that further studies in this direction will strongly benefit from: (i) considering the context-dependent functional role of m6A, (ii) profiling m6A patterning using both base-resolution (miCLIP [25]) and more quantitative methods (m6A-LAIC-seq [26]), (iii) characterizing the impact of m6A on the life-cycle of RNA polymerase II complexes [11]. These approaches will be critical to fully decipher the impact of this and other RNA modifications on the fate of coding and noncoding RNA species.

## Figures and Tables

**Figure 1 genes-10-00028-f001:**
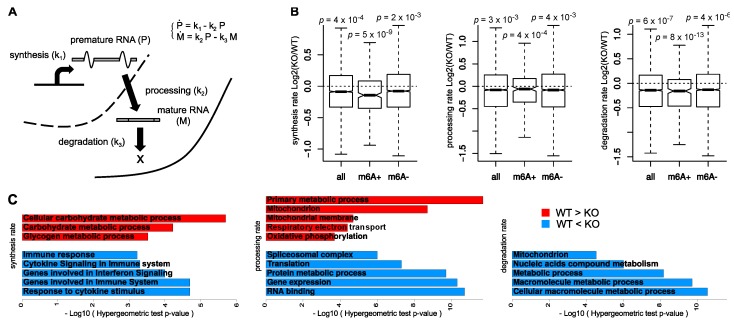
(**A**) Quantification of the kinetic rates of RNA synthesis, processing, and degradation through mathematical modeling of the RNA life-cycle. (**B**) Distributions of changes in the kinetic rates between untreated wild type (WT) and knock-out (KO) cells for all modeled transcripts and for the subsets of mA+ and m6A− transcripts; Wilcoxon *p*-values testing a negative shift of each distribution are reported. (**C**) GeneOntology (GO) functional enrichment analysis for the top-ranking differential genes for each kinetic rate.

**Figure 2 genes-10-00028-f002:**
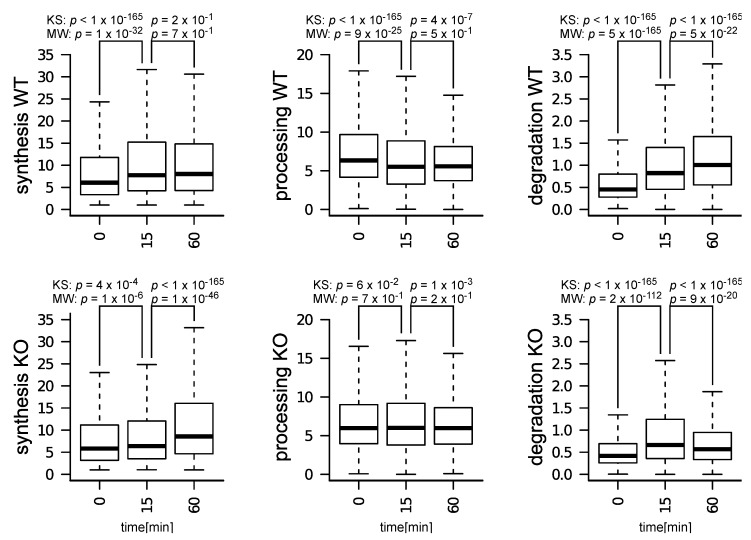
Distributions of synthesis, processing, and degradation rates in WT and KO cells, at: 0, 15, and 60 min after IL-7 induction. The figure shows *p*-values resulting from the application of the Kolmogorov–Smirnov (KS) and of the Mann–Whitney (MW) tests on each pair of distributions tested.

**Figure 3 genes-10-00028-f003:**
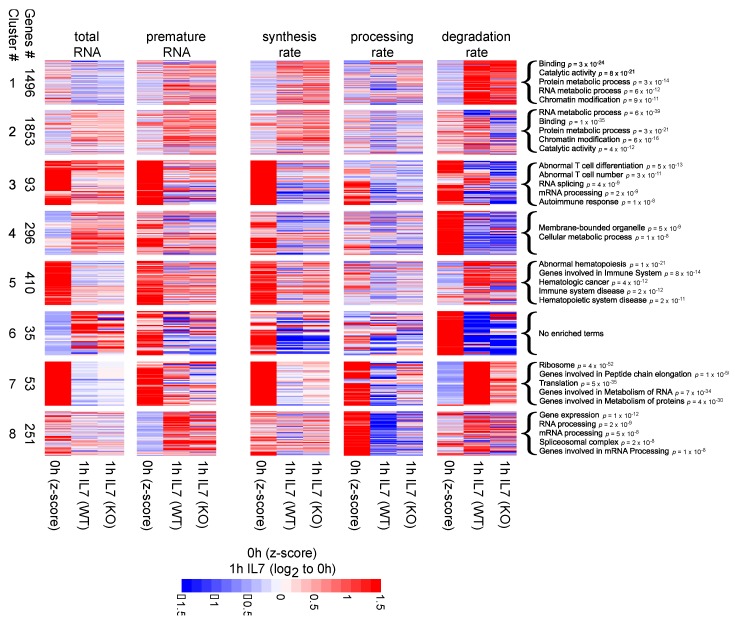
Heatmaps showing the 8 largest clusters emerging from our analysis of Z-scores and Log2 Fold Changes. For each cluster and each quantity among: total RNA, premature RNA, synthesis, processing and degradation rates, the figure reports: the average Z-score between WT and KO, the Log2 Fold Change at 60 min for WT and the Log2 fold change at 60 min for KO. Enriched GO terms are indicated for each cluster.

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
