# Peer review of "m6A-Dependent RNA Dynamics in T Cell Differentiation"

_genes, 2019, doi:10.3390/genes10010028_

Round 1

Reviewer 1 Report

In this manuscript, Furlan et al. reanalyzed the RNA-seq data originally published in Nature (Li et al., 2017) using the R package INSPEcT previously developed by the authors and reported that METTL3-mediated m6A modification impacts not only on RNA stability but also on RNA synthesis/processing during IL-7 induced T cell differentiation in mouse. Although I raised some points that would improve the manuscript, the results in the current work are informative for many researchers in the field. My comments are listed below.

1. The authors should cite and discuss the conclusion made by Ke et al. (Genes Dev 2017). In the paper, they used Mettl3 KO cells and concluded that m6A is not required for most splicing.  

2. The authors may consider citing some additional recent evidences suggesting the importance of m6A for RNA processing (e.g. Bartosovic et al., NAR 2017; Louloupi et al., Cell Rep 2018; Kasowitz et al., PLoS Genet 2018). 

3. Figures 1 and 2: Y-axis labels should be indicated. Also, violin plots with box inside or at least notched box plots will provide more information regarding the distributions.

4. Lines 162-175: It would be nice if the authors could show the summary of functional enrichment analysis in a figure.

5. Lines 208-254:  There are no description about Clusters 4 and 6 in the manuscript. Can the authors discuss the characteristics of these clusters?

6. Showing the summarized information for each cluster (number of genes, significantly enriched ontology terms with p-values, chractersitics (stability, expression level and processing rate--high or low)) in Figure 3 alongside the heatmaps would help readers to follow the results more easily.

Author Response

Reviewer #1

In this manuscript, Furlan et al. reanalyzed the RNA-seq data originally published in Nature (Li et al., 2017) using the R package INSPEcT previously developed by the authors and reported that METTL3-mediated m6A modification impacts not only on RNA stability but also on RNA synthesis/processing during IL-7 induced T cell differentiation in mouse. Although I raised some points that would improve the manuscript, the results in the current work are informative for many researchers in the field. My comments are listed below.

We would like to thank the Reviewer for the overall positive considerations on the manuscript.

1. The authors should cite and discuss the conclusion made by Ke et al. (Genes Dev 2017). In the paper, they used Mettl3 KO cells and concluded that m6A is not required for most splicing.
2. The authors may consider citing some additional recent evidences suggesting the importance of m6A for RNA processing (e.g. Bartosovic et al., NAR 2017; Louloupi et al., Cell Rep 2018; Kasowitz et al., PLoS Genet 2018).

We thank the Reviewer for the opportunity to clarify this. We recognize that previous studies were poorly cited in the manuscript. We revised introduction and discussion incorporating references about m6A and co-transcriptional regulation, including its deposition within premature RNA, the role of RNAPII elongation, and positive and negative evidence for an involvement in splicing regulation. We also stress the difference between regulation of splicing and regulation of processing dynamics, which do not imply each other.

3. Figures 1 and 2: Y-axis labels should be indicated. Also, violin plots with box inside or at least notched box plots will provide more information regarding the distributions.

We updated all the figures containing boxplots to include notches.

4. Lines 162-175: It would be nice if the authors could show the summary of functional enrichment analysis in a figure.

Results from functional enrichment analyses were added in Figures 1

5. Lines 208-254: There are no description about Clusters 4 and 6 in the manuscript. Can the authors discuss the characteristics of these clusters?

Clusters 4 and 6 were not discussed since they were relatively small, and included poorly expressed genes without strong functional enrichment.

6. Showing the summarized information for each cluster (number of genes, significantly enriched ontology terms with p-values, chractersitics (stability, expression level and processing rate--high or low)) in Figure 3 alongside the heatmaps would help readers to follow the results more easily.

Results from functional enrichment analyses were added in Figure 3, together with the number of genes. Abundance (at time 0h) and variation (following IL7 treatment) for both RNA species and the kinetic rates are already included in the figure.

Reviewer 2 Report

In this manuscript by Mattia Furlan et al., the authors examine the impact of m6A on the RNA synthesis, processing, and degradation. By reanalyzing the omics data of a recently published study, they were able to infer how METTL3 depletion affects the RNA synthesis, processing, and degradation rates. To perform such a study, they took advantage of their in-house developed tool, INSPEcT.

Overall, I find the conclusions of this manuscript interesting and to be potentially impactful to the scientific community. Specifically, there are two novel findings in the current manuscript. First of all, m6A affects in a different way different stages of the RNA life-cycle. Secondly, cells without METTL3, respond differently to IL-7 treatment by modulating the RNA synthesis and degradation rates.

Nevertheless, the study has one major inconsistency and a few major concerns that I pointed out below:

Even though the synthesis and processing rates of m6A-mRNAs are not extensively explored by the current literature, the authors wrongly suggest that the impact of m6A on transcription and co-transcription regulation is far from being established. It is a huge inaccuracy. I would suggest to mention and discuss the paper published by the Darnell group (Shengdong Ke et al (Genes and Development 2017)).  In this paper, m6A has been extensively studied in the context of the nascent transcript. It is mentioned how its addition in the nascent transcript is a determinant of cytoplasmic mRNA stability. Additionally, the authors do not mention the work published by the Agami’s group. Boris Slobodin et al reported how dynamics of RNA polymerase II impact the deposition of m6A on mRNAs. In the same paper, Boris Slobodin et al show how RNA pol II interacts with METTL3. Overall, the authors should add and report this part of the literature.  

It is interesting that the RNA synthesis rate in WT and KO cells did not show any significant difference, but the processing and degradation rates did. It is possible that the authors are underestimating the possibility of a change in the synthesis, processing and degradation rates by looking at the overall population of genes in WT and METTL3-KO cells. The KO should affect only a specific gene category, the m6A-genes. In the original study, authors performed RNAseq in WT and KO cell lines, but they also performed m6A-RIP-seq in WT cells. Can the authors see differences in the distribution of the synthesis, processing and degradation rates for the m6A-containing genes only? Can the authors compare m6A-containing genes versus non-m6A containing genes? It would be interesting to have a similar analysis (comparison of m6A versus non-m6A genes) for the data presented in Figure 2.

In Figure 3, why did the author focus on the log2 to 0h only? Given that the degradation rate in KO cells is substantially impacted at 15 minutes, it would be interesting to see which genes are affected after 15 minutes at least in the KO cells.

“In the context of T cells homeostasis, m6A depletion causes a global slowdown in both processing and degradation dynamics, while leaving synthesis rates unaffected.”

How can the author explain the absence of a change in the synthesis rate given that methylation is considered to occur co-transcriptionally?

Moreover, I would like to point out a few minor points:

It would help the reader to introduce the INSPEcT terminology. Thus, a simple summary of what RNA synthesis, processing, and degradation rates are would simplify the understanding of the collected results.

215: Consider to switch the term “half-lives” with degradation rate. Also in Figure 3, it would help the reader to define the directionality of the log2 FC. For instance, in the figure legend, it would help to mention that a positive log2 FC suggests higher synthesis rate but lower degradation rate. 

Author Response

Reviewer #2

In this manuscript by Mattia Furlan et al., the authors examine the impact of m6A on the RNA synthesis, processing, and degradation. By reanalyzing the omics data of a recently published study, they were able to infer how METTL3 depletion affects the RNA synthesis, processing, and degradation rates. To perform such a study, they took advantage of their in-house developed tool, INSPEcT.

Overall, I find the conclusions of this manuscript interesting and to be potentially impactful to the scientific community. Specifically, there are two novel findings in the current manuscript. First of all, m6A affects in a different way different stages of the RNA life-cycle. Secondly, cells without METTL3, respond differently to IL-7 treatment by modulating the RNA synthesis and degradation rates.

We would like to thank the Reviewer for the overall positive considerations on the manuscript.

Nevertheless, the study has one major inconsistency and a few major concerns that I pointed out below:

Even though the synthesis and processing rates of m6A-mRNAs are not extensively explored by the current literature, the authors wrongly suggest that the impact of m6A on transcription and co-transcription regulation is far from being established. It is a huge inaccuracy. I would suggest to mention and discuss the paper published by the Darnell group (Shengdong Ke et al (Genes and Development 2017)). In this paper, m6A has been extensively studied in the context of the nascent transcript. It is mentioned how its addition in the nascent transcript is a determinant of cytoplasmic mRNA stability. Additionally, the authors do not mention the work published by the Agami’s group. Boris Slobodin et al reported how dynamics of RNA polymerase II impact the deposition of m6A on mRNAs. In the same paper, Boris Slobodin et al show how RNA pol II interacts with METTL3. Overall, the authors should add and report this part of the literature.

We thank the Reviewer for the opportunity to clarify this. We recognize that previous studies were poorly cited in the manuscript. We revised introduction and discussion incorporating references about m6A and co-transcriptional regulation, including its deposition within premature RNA, the role of RNAPII elongation, and positive and negative evidence for an involvement in splicing regulation. We also stress the difference between regulation of splicing and regulation of processing dynamics, which do not imply each other.

It is interesting that the RNA synthesis rate in WT and KO cells did not show any significant difference, but the processing and degradation rates did. It is possible that the authors are underestimating the possibility of a change in the synthesis, processing and degradation rates by looking at the overall population of genes in WT and METTL3-KO cells. The KO should affect only a specific gene category, the m6A- genes. In the original study, authors performed RNAseq in WT and KO cell lines, but they also performed m6A-RIP-seq in WT cells. Can the authors see differences in the distribution of the synthesis, processing and degradation rates for the m6A-containing genes only? Can the authors compare m6A-containing genes versus non-m6A

containing genes? It would be interesting to have a similar analysis (comparison of m6A versus non-m6A genes) for the data presented in Figure 2.

We agree that stratifying our observations for m6A+ and m6A- transcripts would be interesting. However, we would like to stress that our conclusions did not aim at distinguishing between direct and indirect effects of m6A (and its modulation), due to lack of data. Indeed: (i) m6A profiling within this study is only available in the untreated WT condition, and (ii) there is 28% residual m6A in KO cells.

Nevertheless, we tried comparing the rates distributions in WT and KO untreated cells. First, we evaluated the effect of Mettl3 KO by plotting the ratios of each kinetic rate in KO vs WT cells. Compared to the previous analysis, this revealed that all kinetic rates, including synthesis, are reduced in m6A-depleted cells (Figure 1B). Second, as reported in the same panel, all kinetic rates are also reduced in KO compared to WT cells when focusing on m6A+ transcripts, in agreement with the global analysis (the “all” category in the figure). However, a similar while weaker and less significant trend could also be observed for m6A- transcripts.

Summarizing, the previously described global effect was now expanded to include the rate of synthesis. This effect could be recapitulated in the population of m6A+ transcripts. However, it remains controversial whether it is a direct or indirect effect of m6A, given the difficulty in defining m6A- transcripts (a good amount of false negatives are expected). Noteworthy, the marked involvement of genes related to RNA processing (Figure 1C and clusters 1,2,3,7 and 8 in Figure 3) might be in agreement with a hypothetical indirect effect of m6A on processing dynamics. The manuscript was edited including these new data and the corresponding discussion.

Regarding Figure 2 (the comparison of WT vs KO cells after IL7 treatment), global trends could be confirmed on the subset of m6A+ genes. Nevertheless, a similar picture emerges also when considering m6A- genes (see the figure below). Altogether, this raises again the point of direct vs indirect effects. Eventually, we believe that this is too speculative, and we do not feel confident on including these results in the manuscript, due to residual m6A in the KO and due to the lack of m6A profiling in the untreated KO, and in both WT and KO following IL7 treatment.

In Figure 3, why did the author focus on the log2 to 0h only? Given that the degradation rate in KO cells is substantially impacted at 15 minutes, it would be interesting to see which genes are affected after 15 minutes at least in the KO cells.

As discussed in the methods, the clustering reported in Figure 3 is obtained using data for the whole time course, including the 15’ time point, which are integrally reported in Supplemental Figure 4. To facilitate the discussion, and due to the similarity of the response at 15’ (slightly weaker) and 60’ (stronger), the heatmap was simplified in Figure 3, including only 60 minutes vs the untreated condition.

“In the context of T cells homeostasis, m6A depletion causes a global slowdown in both processing and degradation dynamics, while leaving synthesis rates unaffected.” How can the author explain the absence of a change in the synthesis rate given that methylation is considered to occur co-transcriptionally?

As discussed above, during this revision we reconsidered the impact of m6A on RNA synthesis in the untreated condition. We now report that all kinetic rates are impacted by m6A (see above).

Moreover, I would like to point out a few minor points:

It would help the reader to introduce the INSPEcT terminology. Thus, a simple summary of what RNA synthesis, processing, and degradation rates are would simplify the understanding of the collected results.

A figure depicting the RNA life cycle and how it is modeled to quantify the kinetic rates was included (Figure 1A).

215: Consider to switch the term “half-lives” with degradation rate. Also in Figure 3, it would help the reader to define the directionality of the log2 FC. For instance, in the figure legend, it would help to mention that a positive log2 FC suggests higher synthesis rate but lower degradation rate.

While RNA half-life is probably more commonly used than RNA degradation, we believe that mixing two kinetic rates (synthesis and processing) with the term half- life (with an opposite interpretation) would complicate the reading. We occasionally discuss the interpretation of the degradation rates in the sense of half-life in the text.

Round 2

Reviewer 2 Report

Overall, I still consider this manuscript interesting and to be potentially impactful to the scientific community. It opens a new perspective on the role of m6A especially during the synthesis process which is unexplored yet. 

In particular, I appreciate that the authors tried to stratify the differences for m6A and non-m6A transcripts. I found very interesting to show that the m6A transcripts are more affected in all-kinetic rates. However, I would suggest emphasizing less the change in the processing step. Indeed, while the trend for the synthesis and degradation rate is more than obvious, the processing step doesn't seem to be affected at the same extent. Please, consider mentioning that in the text.